# Is Botulinum Toxin Effective in Treating Orofacial Neuropathic Pain Disorders? A Systematic Review

**DOI:** 10.3390/toxins15090541

**Published:** 2023-09-01

**Authors:** Matteo Val, Robert Delcanho, Marco Ferrari, Luca Guarda Nardini, Daniele Manfredini

**Affiliations:** 1Department of Biomedical Technologies, School of Dental Medicine, University of Siena, 53100 Siena, Italy; 2School of Dentistry, University of Western Australia, Perth 6009, Australia; 3Unit of Oral and Maxillofacial Surgery, Ca’Foncello Hospital, 31100 Treviso, Italy

**Keywords:** botulinum toxin type A, orofacial neuropathic disorders, essential trigeminal neuralgia, pain, quality of life

## Abstract

Background: The aim of this paper is to provide a systematic review of the literature regarding the clinical use of botulinum toxin (BTX) to treat various orofacial neuropathic pain disorders (NP). Methods: A comprehensive literature search was conducted using Medline, Web of Science, and the Cochrane Library databases. Only randomized clinical trials (RCT) published between 2003 and the end of June 2023, investigating the use of BTX to treat NP, were selected. PICO guidelines were used to select and tabulate the articles. Results: A total of 6 RCTs were selected. Five articles used BTX injections to treat classical trigeminal neuralgia, and one to treat post-herpetic neuralgia. A total of 795 patients received BTX injections. The selected studies utilised different doses and methods of injections and doses. All the selected studies concluded superiority of BTX injections over placebo for reducing pain levels, and 5 out 6 of them highlighted an improvement in the patient’s quality of life. Most of the studies reported transient and mild side effects. Conclusion: There is evidence of the efficacy of BTX injections in orofacial pain management. However, improved study protocols are required to provide direction for the clinical use of BTX to treat various orofacial neuropathic pain disorders.

## 1. Introduction

Pain is an unpleasant sensory and emotional experience associated with or resembling that associated with actual or potential tissue damage [1]. Specifically, neuropathic pain is caused by an injury or a disease of the somatosensory nervous system. Neuropathic pain is divided into central (pain caused by a lesion or disease of the central somatosensory nervous system) and peripheral (a disease or trauma of the peripheral somatosensory nervous system) [2].

The appropriate management of neuropathic pain can be challenging and, as with other chronic pain conditions, should be approached within a biopsychosocial framework. Pharmacological management is considered a component within an overall approach to improving patients’ quality of life and function [3]. Based on systematic reviews and meta-analysis of randomized controlled clinical trials, a number of published guidelines for pharmacological treatment of neuropathic pain are available [4].

Carbamazepine and oxcarbazepine are considered first line pharmacological treatments for cranial nerve neuralgias. For other neuropathic pain disorders, to date, the highest quality evidence exists for tricyclic antidepressants, pregabalin, gabapentin, and serotonin and noradrenaline reuptake inhibitors. Tramadol, topical lidocaine, and topical capsaicin could be considered as second choices. Stronger opioids, including tapentadol, could be considered third line but are generally not considered suitable for non-cancer pain. It should be remembered that opioids and gabapentinoids have abuse potential [5].

The use of botulinum toxin type A (BTX) has gained increasing attention and importance in the management of peripheral neuropathic pain. Studies have demonstrated a moderate efficacy with reduced side effects as compared to systemically administered medication [6]. BTX is a neurotoxin derived from Clostridium botulinum and causes relaxation of skeletal muscles by inhibiting the release of acetylcholine at neuromuscular junctions [7]. BTX also has been demonstrated to inhibit the release of substance P, glutamate, and calcitonin gene-related peptide (CGRP) from sensory neurons [8]. This decreases pain perception by inhibiting both peripheral and central nervous system pain pathways [9]. A systematic review [10] showed a significant benefit of BTX injections in patients affected by neuropathic pain due to postherpetic neuralgia [11,12], spinal cord Injury [13,14], peripheral nerve lesion [15], diabetic neuropathy [16,17,18,19], post-traumatic/postoperative neuropathies [20], myofascial pain [21,22], and carpal tunnel syndrome [23]. Furthermore, numerous papers have demonstrated that BTX intervenes on the modulation of pain perception in the orofacial district [24,25,26,27,28]. Another study [29] highlighted the efficacy of BTX in the treatment of a patients suffering from hemifacial spasms and trigeminal neuralgia.

Although numerous studies have been published evaluating the efficacy of BTX in the management of various neuropathic pain disorders, there remains a paucity of publications specifically addressing the use of BTX for the management of orofacial neuropathic pain disorders, and any findings have not been systematically summarized. Within these premises, the aim of this paper is to systematically review the scientific evidence regarding the safety and efficacy of BTX in the management of orofacial neuropathic pain disorders which have emerged from published randomized controlled trials.

## 2. Materials and Methods

### 2.1. Search Strategy and Criteria for Selecting Articles and Registration

A systematic review of the literature addressing the use of BTX in the management of neuropathic pain affecting the orofacial area (NP) was carried out via a literature search of the MedLine and ISI Web of Sciences databases and the Cochrane Library, following the PRISMA guidelines [30]. Referenced studies within reviewed articles were also included if they met our inclusion criteria. The Pub Med database search utilized a combination of different search keywords (see Appendix A for details): botulinum toxin neuropathic pain, botulinum toxin neuropathy, botulinum toxin neuralgia, botulinum toxin burning mouth, botulinum toxin neurology, botulinum toxin glossopharyngeal neuralgia, botulinum toxin trigeminal neuralgia, botulinum toxin auriculotemporal neuralgia, botulinum toxin postherpetic, and botulinum toxin pain. This systematic review was registered, and the protocol is available on the National Institute for Health Research International prospective register of systematic reviews, PROSPERO, with the number CRD42023403119.

### 2.2. Inclusion Criteria

The inclusion criteria included randomized controlled clinical trials (RCTs) investigating the use of BTX to treat neuropathic pain in the orofacial region. In addition, studies had to be written in English.

### 2.3. Exclusion Criteria

The following publications were omitted: non-randomized controlled trials, non-controlled trials, systematic reviews or meta-analyses, non-systematic reviews, case reports, studies not reporting the use of BTX in NP, studies reporting data from previous publications, opinion papers, letters to the editor, and articles published before year 2000.

### 2.4. Selection of Participants

The participants included adults of both genders diagnosed with any type of neuropathic disorders/neuralgia involving the orofacial area (trigeminal neuralgia/neuropathy, auriculotemporal neuralgia/neuropathy, postherpetic neuralgia/neuropathy, nervus intermedius neuralgia/neuropathy, burning mouth syndrome, peripheral painful traumatic trigeminal neuropathy, persistent idiopathic facial pain, and central post-stroke pain) who have undergone treatment with BTX.

### 2.5. Methods

The review process initially addressed the titles and abstracts (TiAb screening) and then the full-text papers. Two different reviewers (MV, DM) separately performed the process and discussed the differences. The full text of the selected articles that met the eligibility criteria were retrieved and reviewed in-depth by two reviewers (RD, LGN). In each study, the following data were extracted: author (s), year of publication, study design, sample size, gender and age of participants, follow-up period, outcome variables, and results. A PICO-like [31] structured reading (i.e., BP^—patients/problem/population, BI^—intervention, BC^—comparison and BO^—outcome) was adopted, if possible, based on the following question: In patients with various NDs (P), do BTX injections (I), as compared versus other treatments (C), reduced pain levels and improve function (O)? A descriptive analysis was then conducted on the selected studies. As the timeline in which the articles were selected was very wide, it was impossible to structure the review according to PICOT (in which T stands for “time frame”).

### 2.6. Statistical Analysis

The intent was to perform a meta-analysis for this systematic review; however, due to the marked heterogeneity of the studies, this proved impossible. A descriptive analysis of the studies was performed.

### 2.7. Quality of RCT Selected

Grading of the level of evidence was based on the work of Sackett and colleagues and is summarized in Figure 1 [32]. The Jadad score (Figure 2) was used to assess the quality of the double blinding, randomization, and flow of patients. The scores ranged from 0 (bad) to 5 (good). Based on these, the overall quality of the methods was assessed.

### 2.8. Outcome

Primary Outcomes of the reviewed studies were:Pain levels were evaluated using a reliable, validated scale, like the visual analogue scale (VAS);If a validated tool was available, health-related quality of life was assessed;The percentage of participants who experienced major adverse events were measured, including life-threatening situations, hospitalization, or incidents that caused serious disability or incapacity (e.g., infection, dysphagia);Participants with at least 1 adverse event (e.g., hypersensitivity reactions such as anaphylaxis, urticaria, soft tissue oedema, dyspnea, or allergic reaction) were evaluated.

Secondary Outcomes were:Function was assessed by a validated questionnaire;Utilisation of analgesic medication was assessed by type and dose used per day.

## 3. Results

A total of 795 papers were found. Six (*n* = 6) randomized clinical trials were selected via the keywords searched (see Appendix A). The flowcharts of the article selection process for all the search queries can be found in Figure 3. Due to the wide variability in the study methods and the evaluation of results, a meta-analysis of findings could not be performed. Furthermore, despite the search with specific keywords for orofacial neuropathic pain disorders, the only RCTs that were found were for treatment with botulinum toxin for classical trigeminal neuralgia [33,34,35,36,37] and post herpetic neuralgia [12]. Finally, the psychological aspect of pain perception holds significant value and can potentially impact the study’s results, but none of the studies assessed the psychological condition in the patient’s selection process.

### 3.1. Study Characteristics

Table 1 displays the characteristics of the included studies (published between 2003 and 2023).

They included a total of seven RCTs (*n* = 365 patients) of whom 226 were treated with the injection of BTX. Males outnumbered females (M:F = 192:173). The mean age for both case and control groups ranged between 46.4 (±7.7) and 77.73 (±8.41). There was no significant difference between BTX-A and the control groups in terms of frequency of attacks and pain severity before treatment. All selected studies had a placebo (saline injection) control group with the exception of Zhang et al. (2017) [34]. Zhang et al. (2014) [33] compared two groups of patients suffering from trigeminal neuralgia treated with different doses of BTX (25 U BTX group and 75 U BTX group) against a placebo. A later study [34] compared a single dose versus a repeat dose. The single dose group received a first dose of 70–100 U with the second group receiving an initial dose of 50–70 units, which was repeated two weeks later. Finally, Xiao et al. [12] was the only paper that compared BTX with another injected medication (lidocaine) in addition to the placebo group.

### 3.2. Quality of RCTs Selected

Table 2 shows the Jadad score and the level of evidence of the six selected items. The level of evidence is considered high, as all studies are randomized controlled trials (RCTs). However, the study conducted by Zhang et al. (2017) [34], which involved 100 participants, had low quality in terms of double blinding, randomization, and patient flow. On the other hand, Shehata et al. [36], Wu et al. [38], and Xiao et al. [12] had good Jadad scores.

### 3.3. Types, Number of Administrations, Sites and Doses of Botulinum Toxin, and Side-Effects

All studies [12,33,34,35,36,37] evaluated the effectiveness of botulinum toxin type A, and the sites of injection were selected on the basis of subjective pain perception and areas demonstrating evidence of tactile allodynia. The amount of BTX-A injected, injection technique, the site, and the number of injections varied between the studies. The amount of BTX-A injected ranged from a minimum of 25 U to a maximum of 140 U) [34]. The number of injection sites ranged from 8 [36] to 25 [33] depending on the width of the interested area. In each study, the injections were performed at a distance of 10–15 mm from each other. Most of the studies reported transient and mild side effects. The reported side effects were facial asymmetry, weakness, hematoma, oedema at the site of injection, itching, and pain at the site of injection [12,33,34,36,37]. Zuniga and colleagues noted that one participant in the BTX group and five participants in the placebo group experienced hiccups or anaesthesia in the affected area [35]. In addition, three individuals from the BTX group and three patients from the placebo group exhibited a decrease in the corneal reflex.

### 3.4. Period of Follow-Up and Quality of Life and Pain Assessments

The longest period of follow-up was 6 months) [34] while the shortest follow-up time was 8 weeks) [33]. There was lack of homogeneity in the choice of quality of life assessment questionnaires in the various studies. Zhang et al. in 2014 [33] evaluated the overall response to treatment based on the Patient Global Impression of Change (PGIC) scale but did not evaluate the improvement in quality of life in his later study. Other quality of life assessment strategies utilized were the [35] Short Form (36) (SF36) [35] and 10-point quality of life (QoL) scale adopted from the American Chronic Pain Association [36],and sleeping time (hours), daily activity, diet, and stance in a day [12].

### 3.5. Effect on Pain Reduction

All the studies selected [12,33,34,35,36,37] used the visual analogue scale (VAS) to evaluate pain.

All studies included in the review demonstrated a statistically significant effect of BTX injections reducing pain intensity compared to placebo. The onset of analgesic effects averaged between 7 [12,33] and 15 [36,37] days, lasting up to 3 months [12,35]. Studies comparing different doses of BTX failed to demonstrate a significant difference in pain relief at higher doses. Table 3 summarizes the results obtained in pain management.

### 3.6. Quality of Life

Most of all the selected works [12,33,34,36,37], despite using different methods, demonstrated improved quality of life of for patients treated with BTX.

In the management of trigeminal neuralgia, Shehata et al. [36] showed a significant decrease in the number of weekly rescue medications and an increase in the QoL functioning scale following BTX injections. Two studies utilizing the PGIC found that patients who received BTX had a “great improvement” or “very much improvement” in pain symptoms compared with the placebo group (*p* < 0.01 in both studies). Moreover, Zhang et al. (2014) [33] failed to find a significant difference between lower and higher BTX doses (25 U and 75 U) in the improvement of PGIC scores (*p* > 0.05). However, using the SF36, Zuniga et al. [35] failed to find an improvement in quality of life following BTX administration.

Xiao et al. [12] recorded sleep time (hours) to assess any improvement in the quality of life of patients suffering from post-herpetic neuralgia. Over the course of three months post-treatment, the amount of time each group slept consistently increased from day one (*p* < 0.01). The amount of time spent sleeping was significantly increased with the use of BTX-A if compared with the other two group (injected with lidocaine and placebo) (*p* < 0.01).

## 4. Discussion

The treatment of neuropathies/neuralgias in the orofacial region represents a clinical challenge for both patients and health care providers. Orally administered medications (and in particular carbamazepine for trigeminal neuralgia) are considered the gold standard in the treatment of NP; however, these medications can be poorly tolerated and associated with severe side effects. Surgical therapies such as microvascular decompression are invasive and with associated morbidity and even mortality. If an alternative, efficacious treatment with reduced side effects could be established, the treating practitioner will have further management options at their disposal. BTX seems to have these characteristics, and, since 2002 [29], it has been used successfully in the management of trigeminal neuralgia [8,9]. The ability of BTX in reducing pain perception in other orofacial pathologies has been widely demonstrated [24,25,26,27,28].

Unfortunately, using our strict inclusion criteria to analyse the literature, only two orofacial neuropathic pain conditions were identified: classical trigeminal neuralgia [33,34,35,36,37] and post-herpetic neuralgia [12]. Publications addressing the use of BTX to treat other likely orofacial neuropathic pain conditions such as burning mouth disorder, post traumatic trigeminal neuropathic pain, persistent idiopathic facial, and dentoalveolar pain simply failed to meet the criteria used for inclusion in this review. There is also low-quality evidence for use of BTX injections to treat some conditions possibly related to trigeminal neuralgia, such as SUNCT and SUNA. It is evident that additional research, beyond just case reports, is necessary to establish high-quality evidence on the effectiveness of using BTX to treat the aforementioned conditions, as such evidence is currently non-existent. The use of BTX in these situations is therefore lacking a solid scientific evidence base but could be considered clinically on an ‘ad hoc’ basis perhaps where other treatments have failed.

The studies that were included had small sample sizes, with only three out of six studies being carried out with more than 50 participants [12,33,34]. The high cost of botulinum toxin and the fact that the management of neuropathic disorders with BTX is “off-label” may explain the low number of studies and participants. All the studies stated the diagnostic criteria used for the different disorders.

Furthermore, all patients in the selected studies had been treated with anti-epileptic medications (usually carbamazepine or oxcarbazepine, with gabapentin being the next most popular), which were all maintained during the studies. Details of the medications being taken were provided in the studies. However, Xiao et al. [12] suspended all medications and introduced transcutaneous electrical nerve stimulation (TENS) in the management of pain during the trial.

The sites of BTX injections varied between the different articles, with some being directly into trigger zones [33,34,35,36,37] and others within the affected dermatomes/oral mucosa involved [12]. BTX was injected intradermally [33,34,37], submucosally [33,34] or subcutaneously[12,35,36], but the depth of injection in millimetres was never stated.

The paper which compared single different dosages of BTX [33] and a second dose two weeks later [34] found no statistical difference in VAS scores where a higher dose of BTX was used. Higher doses were found to be safe.

All the studies demonstrated that BTX had a statistically significant efficacy in pain management compared to placebo [12,33,35,36,37] and lidocaine [12]. Furthermore, all the papers included in this review stated that the different indices for quality of life assessment are significantly improved compared to placebo. Due to these findings and the fact that each study utilised different dosages from very low (25 U [33]) to very high (140 U [34]), it was not possible to identify a common or recommended protocol for the management of orofacial neuralgia with botulinum toxin. It is unclear from the included studies which patients may best benefit from using BTX injections. There is weak evidence that BTX further reduces pain intensity when used as an adjunct to the commonly used anti-epileptic drugs [39].

BTX could prove to be exceptionally beneficial for elderly patients who are unable to bear the adverse effects of medication and may not be apt for, or apprehensive of, severe complications from microvascular decompression surgery [38]. It may have use as a “rescue” strategy for acute exacerbations of primary trigeminal neuralgia [40]. In doses of 100–300 U, there is evidence of BTX efficacy to treat localized peripheral neuropathic pain. This may be particularly helpful for patients who exhibit allodynia, limited thermal deficit, and have residual intraepidermal nerve fibres as determined by a skin punch biopsy [14,41].

The evidence from this systematic review suggests that botulinum toxin type A (BTX-A), when compared to placebo, probably has a clinically significant benefit in the treatment of the orofacial neuropathic pain disorders, trigeminal neuralgia, and post herpetic neuralgia. The overall outcomes consistently favoured BTX-A compared with placebo across studies, while also considering that systemic side effects were minimal and transient. It should be noted that different diagnostic criteria were used for patient selection (tIHS [36], ICHD-2 [33,34,37] and the classification of chronic pain [35]) and the differences between the infiltration protocols do not make these studies directly comparable.

## 5. Conclusions

Previous reviews of the literature [24,42] have demonstrated evidence of the efficacy of BTX injections in pain management. However, in order to provide a protocol for the tailored use of BTX to treat different orofacial neuropathic pain conditions, improved study protocols, e.g., patient selection, phenotyping, injection techniques, dosing, intervals between doses, etc., and an increased homogeneity of research protocols are required. Furthermore, studies utilizing BTX to treat other presumed orofacial neuropathic conditions are required before recommendations for routine clinical use can be made.

## Figures and Tables

**Figure 1 toxins-15-00541-f001:**
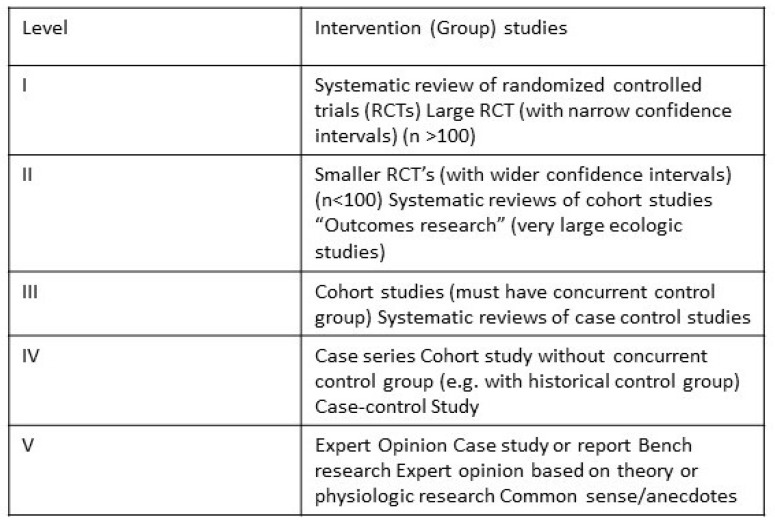
Grading of the level of evidence is based on the work of Sackett and colleagues.

**Figure 2 toxins-15-00541-f002:**
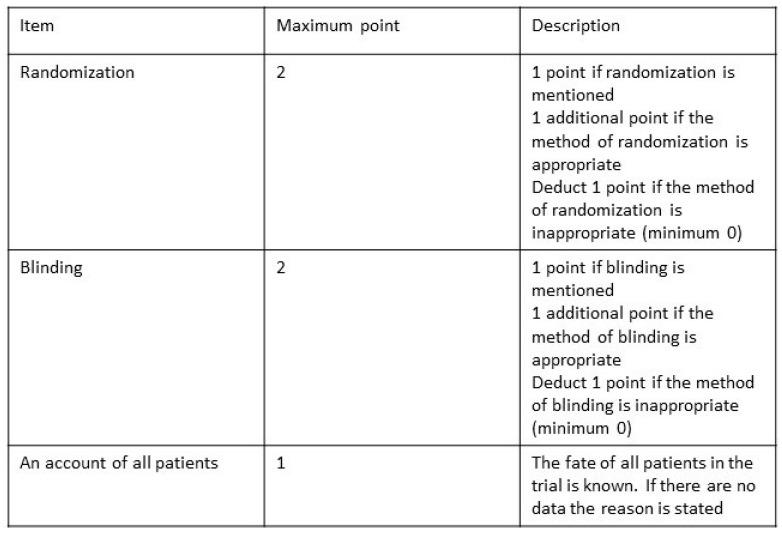
Jadad Scale.

**Figure 3 toxins-15-00541-f003:**
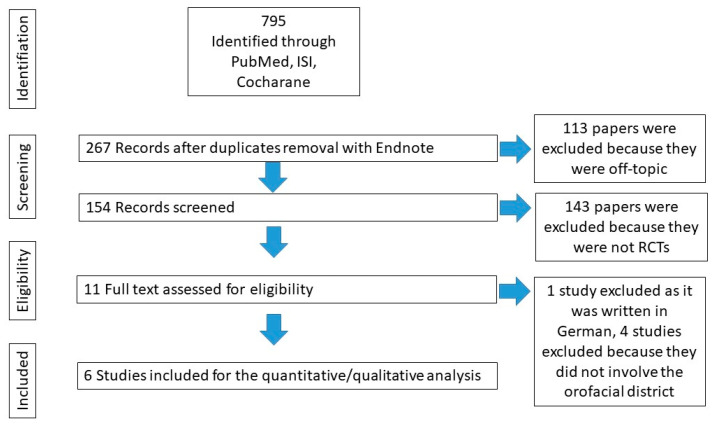
Flowchart that highlights the process of article selection.

**Table 1 toxins-15-00541-t001:** Characteristics of the studies related to the interventions.

Study	Year	Diagnosis	Sample Size (Test/Control)	Test/Control	Site of Injection	Repeated Injection	Weeks between First and Second Injection	Follow-Up	Drop-Out
Shehata [36]	2013	Trigeminal Neuralgia	20 (BTX 10: saline10)	BTX-A (Botox^®^) (100 U Botox in 2 mL preservative-free normal saline, resulting in a concentration of 5 units/0.1 mL) or placebo (2 mL 0.9% NaCl).	“follow the pain” method. If the branch of the mandible is affected, a higher amount of toxin is injected at the back of the masseter muscle to prevent any unwanted cosmetic outcomes.	No	0	12 weeks	0
Wu [37]	2012	Trigeminal Neuralgia	42 (BTX 22: saline 20)	intradermal and/or submucosal injection of BTX-A (75 U/1.5 mL;) or saline (1.5 mL)	“follow the pain” method.	No	0	12 weeks	2
Xiao [12]	2010	Postherpetic neuralgia	60 (20:20:20)	The lidocaine group (0.5% lidocaine) acted as an active control. Non-preserved saline (0.9%) was used for the placebo group. For the BTX-A group, aliquots of 100 IU/vial of BTX-A were reconstituted with 20 mL of saline solution.	Subjects received subcutaneous injections into the affected area, specifically the area demonstrating evidence of tactile allodynia.	No	0	3 months	4
Zhang [34]	2017	Trigeminal Neuralgia	100 (50:50)	Patients in the single-dose group received a local BTX-A injection of 70 to 100 U. The repeated-dose group received an initial BTX-A injection of 50 to 70 U and then another of equal volume 2 weeks later.	“follow the pain” method	Yes	2 weeks	6 months	19
Zhang [33]	2014	Trigeminal Neuralgia	84 placebo (*n* = 28); BTX-A 25 U (*n* = 27); BTX-A 75 U (*n* = 29)	Each vial contained either active botulinum toxin type A (25 U or 75 U) or matching placebo. All three vials were identical in appearance and were reconstituted with 1 mL saline solution (0.9%). For treatment, 1 mL was drawn from vials, and the injections were administered intradermally and/or submucosally.	“follow the pain” method	No	0	8 weeks	0
Zúñiga [35]	2013	Trigeminal Neuralgia	36 (BTX 20: saline 16)	1 mL 0.9% saline plus 50 U of BTX or only 1 mL of 0.9% saline injected subcutaneously in the affected area.	Among the path of the branch/branches involved, patients with involvement of the third branch of the trigeminal nerve also received intramuscularly either 10 U of BTX or matching placebo in the masseter muscle, ipsilateral to the pain location.	No	0	3 months	5

**Table 2 toxins-15-00541-t002:** Jadad score and the level of evidence of the 8 selected items.

Study	Year	Type of Study	Level of Evidence	Jadad Score
Shehata [36]	2013	RCT double blind	II	5
Wu [37]	2012	RCT double blind	II	4
Xiao [12]	2010	RCT double blind	II	4
Zhang [34]	2017	RCT	I	2
Zhang [33]	2014	RCT double blind	II	2
Zúñiga [35]	2013	RCT double blind	II	3

**Table 3 toxins-15-00541-t003:** Summary of the effect of BTX in pain management.

Study	Year	Results in Pain Management
Shehata [36]	2013	Pain reduction at the 12-week endpoint was significant in the BTX group (*p* < 0.0001). The BTX-A group showed a decrease in VAS scores starting from week 2 and maintained it throughout the follow-up period. Furthermore, a significant reduction in paroxysm frequency was observed in the BTX-A group compared to the placebo group from week 2 onwards (*p* < 0.0001).
Wu [37]	2012	At week 2, BTX-A showed a significant decrease in the average VAS scores compared to the placebo. The results showed that BTX-A was significantly better than the placebo in reducing the frequency of attacks. This effect was noticeable as early as the first week.
Xiao [12]	2010	The VAS pain scores decreased in all three groups at day 7 and 3 months after treatment (*p* < 0.01). The group that received BTX-A had a greater decrease in VAS pain scores compared to the lidocaine and placebo groups, which was more significant at day 7 and 3 months after treatment (*p* < 0.01). Out of the three groups tested, the BTX-A group had the lowest percentage (21.1%) of subjects using opioids after treatment, compared to the lidocaine (52.6%) and placebo (66.7%) groups. This difference was statistically significant (*p* < 0.01).
Zhang [34]	2017	The group that received a single dose of the drug had a noticeably longer effect time (*p* = 0.032). The drug response rates between the single-dose and repeated-dose groups did not show significant differences (*p* > 0.05).
Zhang [33]	2014	During the study, the groups that received doses of 25 U and 75 U experienced a significant reduction in VAS scores compared to the placebo group as early as week 1. Throughout the study, there was no significant difference in VAS scores between the 25 U and 75 U groups. At week 8, the response rates for the 25 U group (70.4%) and 75 U group (86.2%) were significantly higher than the response rate for the placebo group (32.1%). However, there was no significant difference in response rates between the 25 U and 75 U groups.
Zúñiga [35]	2013	After three months of the injection, a noticeable contrast was detected in the average VAS score between individuals who received BTX treatment and those who received placebo treatment (VAS 4.75 vs. 6.94, respectively; *t*-test, *p* = 0.01).

## Data Availability

No new data were created in this study. Data sharing is not applicable to this article.

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
