# Peer review of "Is Botulinum Toxin Effective in Treating Orofacial Neuropathic Pain Disorders? A Systematic Review"

_toxins, 2023, doi:10.3390/toxins15090541_

Round 1

Reviewer 1 Report

The topic is very interesting, considering botulinum toxin increased use in painful conditions, even in a off-label mode. However, the authors reached only two types of neuropathies to include in the revision, so, I believe authors could change sentences like "A systematic review of the literature addressing the use of BTX in the management of different neuropathic pain affecting the orofacial area (NP) was carried out." For something like "A systematic review of the literature addressing the use of BTX in the management of neuropathic pain affecting the orofacial area (NP) was carried out." 

I also suggest changing the order of the figures and tables to follow the methodology and results more reasonably. Start with Fig 1 and Fig 2, which are related to methodology items 2.7. Following, in results, include Fig 3, table 1, 2 and 3. 

Authors can better describe in text the results of table  2.

The selected studies described what brand of botulinum toxin was used? If yes, please include that because this can be related to the relative high variability of doses used in each one, once we know there is a difference between the market brands.

No additional suggestion.

Author Response

Dear Reviewer,

Thank you for your kind response and appreciation of our paper. We have made the changes you requested, mainly fixing the error in the order of the figures and tables. However, according to the author's instructions and the template we downloaded, figures should be inserted in a specific chapter. At the same time, tables should follow the text citation.

The BTX type was already present in line 192 (of the reviewed paper that I attached) in chapter 3.3. Types, Number of administrations, Sites and Doses of botulinum toxin and Side-effects.

Thank you for your time and consideration.

Sincerely,

Reviewer 2 Report

Line 30-31, remove, it does not provide value to the paper.

Line 109-111.  If there was not meta-analysis, explain which analysis was performed.

Improve lines 141-142, is not clear.

Line 158-159. offer a brief summary of the table here.  

Line 229, the purpose of the study is not part of the discussion.

Author Response

Dear Reviewer,

Thank you for your kind response and appreciation of our paper. We have made the changes you requested.

Thank you for your time and consideration.

Reviewer 3 Report

This paper presents a systematic review focused on the clinical utilization of botulinum toxin (BTX) for addressing Orofacial Neuropathic Pain Disorders (NP). Through an extensive exploration of databases including Medline, Web of Science, and Cochrane Library, the review specifically targeted randomized clinical trials (RCTs) published between 2003 and June 2023, which investigated BTX as a treatment for NP. The selection process adhered to PICO guidelines and resulted in the identification of 6 RCTs. Among these, 5 studies employed BTX injections for classical trigeminal neuralgia treatment, and 1 study investigated its use for post-herpetic neuralgia. The collective patient count for BTX injections across these studies was 795, each utilizing varying doses and injection techniques. Overall, the selected trials consistently demonstrated the superiority of BTX injections over placebo in diminishing pain levels, with 5 out of 6 studies noting an enhanced quality of life among patients. Transient and mild side effects were predominantly reported. The review underscores the existing evidence supporting the effectiveness of BTX injections in managing orofacial pain. Nevertheless, the authors emphasize the need for refined study protocols to offer clear guidance on the clinical application of BTX for diverse orofacial neuropathic pain disorders.

Botulinum neurotoxin (BTX) offers a broader range of benefits beyond what was initially recognized. In fact, a comprehensive review has highlighted the significant advantages of using BTX injections for patients dealing with neuropathic pain arising from various conditions such as postherpetic neuralgia, spinal cord injuries, peripheral nerve damage , diabetic neuropathy, post-traumatic or postoperative neuropathies, and carpal tunnel syndrome. Moreover, several research papers have underscored how BTX plays a role in influencing the perception of pain within the orofacial region.

In essence, the impact of BTX extends far beyond its well-known cosmetic applications. This powerful neurotoxin has shown promise in alleviating neuropathic pain stemming from a spectrum of causes, encompassing viral infections like postherpetic neuralgia, traumatic injuries like spinal cord damage, and complications arising from conditions such as diabetes. Additionally, BTX has exhibited its potential in ameliorating pain linked to surgical or traumatic events and even in cases of carpal tunnel syndrome.

However, the paper did not discuss the myofascial pain syndrome as one of the BTX treatment. Please add myofascial pain syndrome with BoNT treatment on the passage of line 53. With the reference of “Guidance to trigger point injection for treating myofascial pain syndrome: Intramuscular neural distribution of the quadratus lumborum” and “Anatomical analysis of the motor endplate zones of the suprascapular nerve to the infraspinatus muscle and its clinical significance in managing pain disorder

The potential of BTX doesn't end there. Emerging studies have further demonstrated its involvement in influencing the perception of pain, particularly in the orofacial domain. This implies that BTX's effects on pain modulation could have a positive impact on individuals suffering from discomfort or pain in the facial area. Consequently, the expanding scope of BTX applications underscores its significance not only in medical contexts but also in enhancing our understanding of pain management mechanisms.

At the line. 17 , please change (,) -> (.).

there is nothing to comment on it

Author Response

Dear Reviewer,

Thank you for your kind response and appreciation of our paper. We have made the changes you requested, in particular adding the two citations in the "Introduction" referring to the use of BTX in myofascial pain.

I fully concur with the application of BTX for pain management and myofascial pain, as stated in the systematic review we released in 2022 titled "Botulinum Toxin for Treating Temporomandibular Disorders: What is the Evidence?". Further research needs to be conducted to enhance our understanding of how BTX operates on both the peripheral and central nervous systems to alleviate pain perception.

Thank you for your time and consideration.

Sincerely,

Round 2

Reviewer 3 Report

Thank you for revision. All good to go. 

They have revised it accordingly.